

# Neuro-controller implementation for the embedded control system for mini-greenhouse

Vasyl Teslyuk[1], Ivan Tsmots[1], Natalia Kryvinska[2], Taras Teslyuk[3], Yurii Opotyak[1], Mariana Seneta[1] and Roman Sydorenko[1]

[1] Department of Automated Control Systems, Lviv Polytechnic National University, Lviv, Ukraine
[2] Department of Information Management and Enterprise Systems, Comenius University in Bratislava, Bratislava, Slovakia
[3] Department of Information Systems and Networks, Lviv Polytechnic National University, Lviv, Ukraine

## ABSTRACT

Control of a certain object can be implemented using different principles, namely, a certain software-implemented algorithm, fuzzy logic, neural networks, *etc*. In recent years, the use of neural networks for applications in control systems has become increasingly popular. However, their implementation in embedded systems requires taking into account their limitations in performance, memory, *etc*. In this article, a neuro-controller for the embedded control system is proposed, which enables the processing of input technological data. A structure for the neuro-controller is proposed, which is based on the modular principle. It ensures rapid improvement of the system during its development. The neuro-controller functioning algorithm and data processing model based on artificial neural networks are developed. The neuro-controller hardware is developed based on the STM32 microcontroller, sensors and actuators, which ensures a low cost of implementation. The artificial neural network is implemented in the form of a software module, which allows us to change the neuro-controller function quickly. As a usage example, we considered STM32-based implementation of the control system for an intelligent mini-greenhouse.

## INTRODUCTION

At the processing of technological data from a variety of sensors in the control system (*Knayer & Kryvinska, 2022*; *Teslyuk et al., 2022*; *Teslyuk et al., 2017*), their loss often occurs, there is a need to process fuzzy data, *etc*. In such situations, it is advisable to use special intellectual tools. Additionally, its implementation in embedded systems requires taking into account their limitations in performance, memory, *etc*. In this article, a method is proposed to process the technological data using a microcontroller with an artificial neural network (ANN) implemented by software. It is clear that for most technical problems such a combination is sufficient in terms of speed, and for real-time systems, the neural

Corresponding author
Mariana Seneta,
mariana.y.seneta@lpnu.ua

network can be implemented at the hardware level (*Kravets & Shymkovych, 2020*; *Mishchuk, Tkachenko & Izonin, 2020*).

To achieve the set goal, it is necessary: to develop the structure and algorithm of the neuro-controller for processing technological data (NPTD) functioning; to develop a model on the basis of an artificial neural network for processing technological data, and; to develop a program and hardware means of NPTD.

The article includes the analysis of existing solutions of the specified problem, the development of the functioning algorithm, and the structure of the neuro-controller, which is based on the modular principle. The built ANN model is developed and tested. The peculiarities of the hardware and software implementation of the neuro-controller are also given.

## RELATED WORKS

Modern industry trends are based on the application of Industry 4.0–full automation of production systems, and integration of new technologies, including artificial intelligence (AI) and machine learning into their production facilities and throughout their operations (*Zaimovic, 2019*; *Oztemel & Gursev, 2020*; *Nascimento et al., 2019*; *Pozzi, Rossi & Secchi, 2023*), the large-scale use of smart systems in various spheres of human activity (*Kim et al., 2022*; *Mazza, Tarchi & Juan, 2022*; *Mbungu, Bansal & Naidoo, 2019*) are aimed to save energy and natural resources.

The above-mentioned concept can be practically realized with the use of modern technologies, methods, and models of computing intelligence. To date, a number of ISA-95, IEC 62264, ANSI/ISA-95 and IEC 62264 standards have been developed that define requirements and features in the development of such systems (*Wally, Huemer & Mazak, 2017*; *International Electrotechnical Commission, 2013*; *International Electrotechnical Commission, 2018*; *International Electrotechnical Commission, 2016*).

As a rule, such systems are multi-level and hierarchical (*Teslyuk et al., 2022*; *Teslyuk et al., 2017*). The level of data collection and management of executive mechanisms is located closest to technological processes and controls technological parameters with the help of sensors. Technological data are fuzzy and unstructured and accordingly their effective processing is possible using artificial neural networks (*Wally, Huemer & Mazak, 2017*; *International Electrotechnical Commission, 2013*; *International Electrotechnical Commission, 2018*). In particular, *Teslyuk et al. (2022)* proposed a device based on the neuro-controller, where an artificial neural network is implemented in the form of a program that controls the operation of the microcontroller and the implemented ANN.

Engineers widely use neuro-controllers in the process of achieving technical tasks, where the problem of processing fuzzy input data arises. In particular, this problem arises in construction (*Chang & Sung, 2019*; *Zizouni et al., 2019*)—for seismic exploration problems; in the process of implementing smart home systems (*Teslyuk et al., 2019*; *Teslyuk et al., 2018*)—for the tasks of protecting the building and processing emergency situations; in materials science (*González-Yero et al., 2021*). *Verginis, Xu & Topcu (2023)* describes a learning-based algorithm for the control of autonomous systems that integrates neural

network-based learning with adaptive control. In research a hybrid controller is introduced (*Abougarair, 2023*), that combines a neural controller with a linear quadratic regulator with feedforward PI controller. This adaptive neuro-controller is trained offline to simulate the PI controller.

The conducted analysis makes it possible to state that the implementation of ANN is possible with two approaches: software (*Chang & Sung, 2019*; *Zizouni et al., 2019*; *Teslyuk et al., 2019*; *Teslyuk et al., 2018*; *González-Yero et al., 2021*) and hardware (*Chang, Martini & Culurciello, 2015*; *Nurvitadhi et al., 2017*). The software approach consists in the software implementation of the ANN, which is stored in the microcontroller memory. This approach is more commonly used in practice and makes it possible to change the parameters of the network model during the operation of the neuro-controller, which is an advantage of this approach. For real-time systems, it is necessary to use hardware implementation of ANN to ensure strict requirements for the performance of the designed system. But at the same time, it will be much more difficult to make changes to the network structure. It is proposed to increase the operation performance of the ANN model taking into account their hardware implementation, using FPGA, CPU, and GPU (*Misra & Saha, 2010*; *Nurvitadhi et al., 2016*; *Ovtcharov et al., 2015*). But such hardware implementation has worse values of both weight and size, and economic parameters.

This article uses the first approach to the implementation of the ANN and storing the program in the memory of a standard microcontroller. This makes it possible to provide the requirements for performance, cost and size parameters in the process of collecting and previous processing technological data.

## MATERIALS & METHODS

### Development of the structure and algorithm of the functioning of an intelligent data collection and processing tool

In general, the structure of the basic intelligent data collection and processing tool includes three main components: a subsystem for collecting data about the environment; a subsystem for processing input technological data; and a subsystem of influence on the studied environment.

In mathematical form, the corresponding structure can be written using the following tuple:

$$Ne_{contr} = \langle M_{sensors}, M_{hard-softwore}, M_{actuators}, M_{ints} \rangle \qquad (1)$$

where $M_{sensors}$ is a set of sensors and detectors; $M_{hard-softwore}$ is a set of hardware and software tools; $M_{actuators}$ is a set of actuators that make it possible to influence the studied environment and $M_{ints}$ is an incidence matrix that allows establishing relationships between sensors, software and hardware, and actuators.

Let us consider an intelligent data processing tool using the example of implementing an intelligent mini-greenhouse management system (*Ma, Li & Yang, 2018*; *Suryawanshi et al., 2018*). The intelligent greenhouse provides maintenance of the microclimate and lighting regime for growing plants according to the specified conditions. We implement

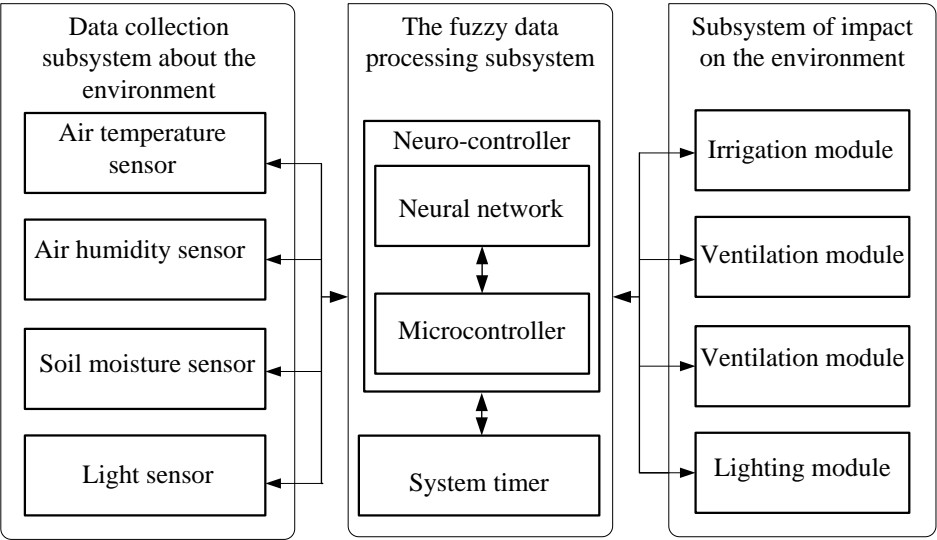

**Figure 1** Structure of the control subsystem of intelligent mini-greenhouse.

the corresponding system on the basis of the developed neuro-controller. The structure of the developed control system consists of two main components (Fig. 1): a microcontroller that runs the main loop of the control program and a neuro-controller software. The control system analyzes input data from various sensors and generates control signals for actuators.

There are four main sensors to gather information about mini-greenhouse:

- air temperature sensor, that monitors the temperature of the environment;
- air humidity sensor to gather information on air humidity;
- sensor to track soil moisture;
- light sensor, that checks outdoor lighting level.

Additionally, a real time clock is used for tracking the time of day.

The block diagram of the algorithm of the neuro-controller for controlling the intelligent mini-greenhouse is shown in Fig. 2.

The above sensors make it possible to determine changes in the mini-greenhouse environment (*Ma, Li & Yang, 2018*; *Suryawanshi et al., 2018*). The actuators were used to influence the medium of the greenhouse. In particular, the developed intelligent greenhouse control system (*Batyuk, Voityshyn & Verhun, 2018*) uses the following actuators(executive modules): the subsystem for watering the soil; the ventilation subsystem to reduce temperature and humidity inside the air control and the cleaning system; the subsystem of heating air; the lighting subsystem that turns on in case of insufficient external light.

For the case of the structure shown in Fig. 1, the set of sensors includes elements, namely:

$$M_{sensors} = (S_1, S_2, S_3, S_4),$$

where $S_1$ is the air temperature sensor; $S_2$ is the air humidity sensor; $S_3$ is the soil moisture sensor; $S_4$ is the illumination sensor of the medium of the mini-greenhouse.

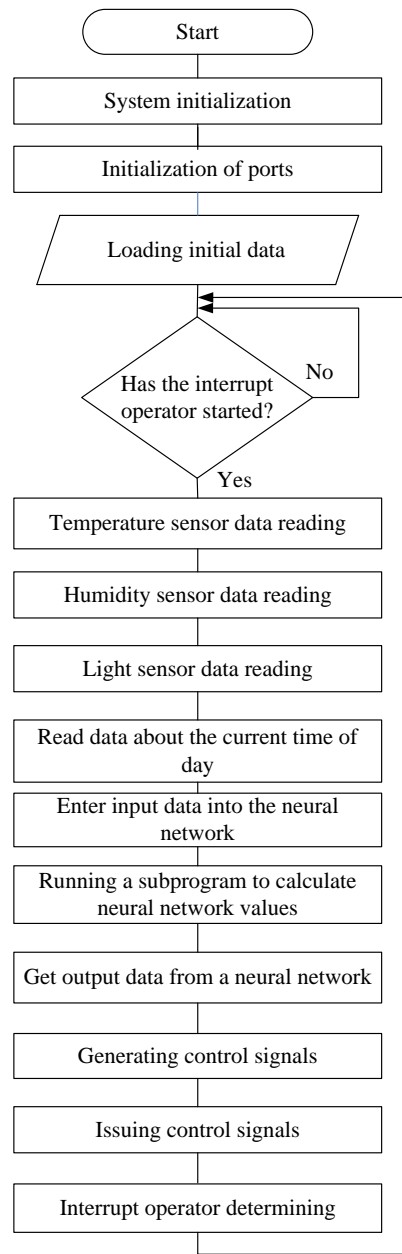

**Figure 2  Block diagram of the neuro-controller algorithm for controlling an intelligent mini-greenhouse.**

A set of hardware and software tools includes only two components: a microcontroller and software that emulates the operation of an artificial neural network. It should be noted that the appropriate set may include several microcontrollers, which are determined by technical and economic feasibility. The set of actuators also includes four elements (modules):

$$M_{actuators} = (A_1, A_2, A_3, A_4),$$

where $A_1$ is a module that implements the function of watering a controlled environment; $A_2$ is a module that provides ventilation of the environment; $A_3$ and $A_4$ heating module and lighting module, respectively.

The incidence matrix for the studied structure of the system in Fig. 1 makes it possible to display the connections between structural elements. It has the following form:

$$M_{int s} = \begin{vmatrix} 1111000000 \\ 0000000000 \\ 0000000000 \\ 0000000000 \\ 1111111111 \\ 0000100000 \\ 0000010000 \\ 0000001000 \\ 0000000100 \\ 0000000010 \\ 0000000001 \end{vmatrix}.$$

The algorithm of the neuro-controller operation includes several basic steps (Fig. 3). The developed algorithm provides an initial step, which is designed to establish initial data (system initialization, port initialization, *etc.*). The following steps are performed sequentially in the cycle: polling and receiving technological data from sensors; the step related to the processing of the received data from sensors by a neural network and the step of forming control signals for the subsystem of influence on the studied environment.

Representation of the structural model of the system in graph form (Eq. (2)) makes it possible to analyze the functioning of the system using existing free software systems.

$$G = (P, I), \tag{2}$$

where $P$ is a set of nodes (components) and $I$ is a set of arcs.

To analyze the operation of the intelligent mini-greenhouse control system, it is advisable to use a structural model in the form of a graph (Fig. 4).

## Development, training and implementation of the artificial neural network model

### Building models of the functioning scenarios of intelligent tools for collecting and processing technological data

Models of system operation scenarios can be described using a number of conditions that are determined by operation modes. Therefore, it is necessary to maintain the temperature

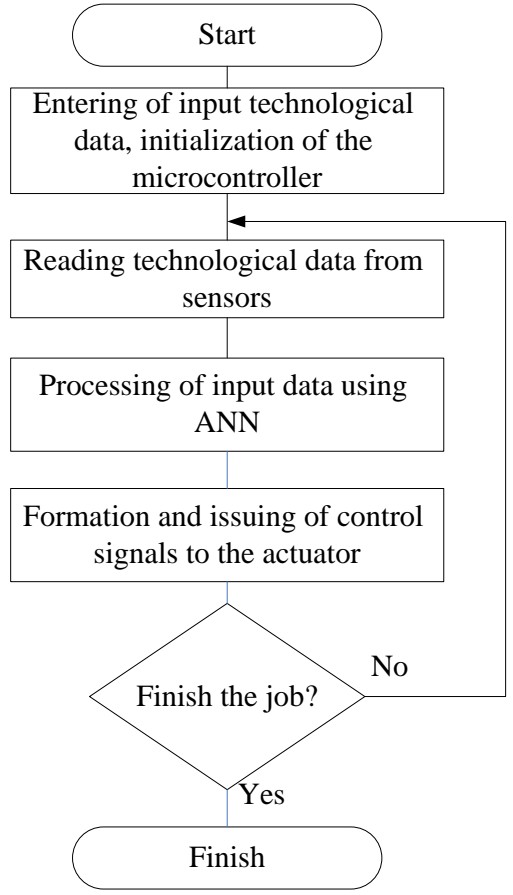

**Figure 3  A simplified block diagram of algorithm of neuro-controller operation.**

regime in the environment of the intelligent mini-greenhouse and the level of illumination during the hours set aside for this. We introduce the following notations, respectively: $T$ is the temperature inside the system; $H_{earth}$ is the soil moisture; $H_{air}$ is an air humidity; $L$ is the level of external lighting; $D$ is the time of day (0–24 h).

During the functioning of the intelligent mini-greenhouse, the following conditions must be maintained:

$$T_{min} < T < T_{max}, H_{earthmin} < H < H_{earthmax},$$

$$H_{airmin} < H < H_{airmax}, L_{min} < L < L_{max}, D_{min} < D < D_{max}, \tag{3}$$

where $T_{min}$, $T_{max}$—minimum and maximum temperature values; $H_{earthmin}$, $H_{earthmax}$—minimum and maximum soil moisture values; $H_{airmin}$, $H_{airmax}$—minimum and maximum values of air humidity; $L_{min}, L_{max}$—minimum and maximum value of the level of exterior light; $D_{min}, D_{max}$—minimum and maximum values of time of day.

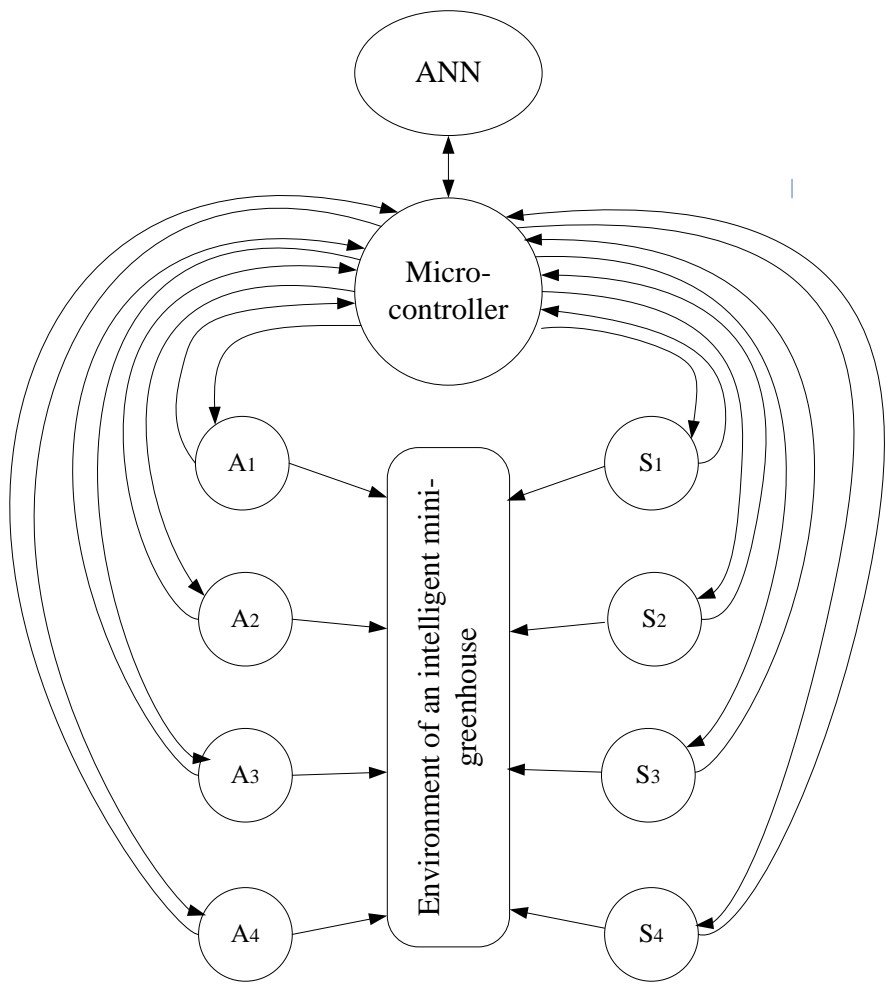

**Figure 4** **Structural model of the intelligent mini-greenhouse control system in the graph form.**

To support the above modes, system operation scenarios that describe the necessary actions to stabilize conditions in the mini-greenhouse environment are developed. Work scenarios include steps that are described and summarized in the Table 1.

Therefore, the ANN receives technological data from the temperature, soil moisture and light sensors. The mode of watering and lighting depends on the time of day. The neuro-controller generates control signals for controlling ventilation shafts, heaters, lighting and watering. Accordingly, the resulting neural network model should have four inputs and four outputs.

## Data preparation for neural network training and testing

To train the neural network, a set of data is generated that describe the received values from the sensors and the expected values for the actuators. For training, it is necessary to generate a sufficient sample that will represent the various states where the system can be in.

**Table 1  Work scenarios of intellectual greenhouse.**

| Meaning of conditions | Action |
| --- | --- |
| High air temperature | Turn on the fan<br>turn off the heater |
| Low air temperature | Turn on the heater<br>turn off the fan |
| High air humidity | Turn on the fan |
| Low air humidity | Turn off the fan |
| High soil moisture | Turn off watering |
| Low soil moisture | Turn on watering |
| High level of outdoor lighting | Turn off the external lighting |
| Low level of external lighting and time for active lighting of plants | Turn on the external lighting |
| Outside the time area for active lighting | Turn off the external lighting |

```
public class Main {
public static void main(String[] args) {
// TODO Auto-generated method stub
GHDataBank bank = new GHDataBank();
List<GHNormalizedState> normalizedData = GHDataNormalizer.normalize(bank.getStates());
printNormalizedDataSet(normalizedData);
}
private static void printNormalizedDataSet(List<GHNormalizedState> normalizedData) {
StringBuilder builder = new StringBuilder();
for(GHNormalizedState state : normalizedData) {
builder.append("\n");
int length = state.getData().length;
for(int i = 0; i < length; i++) {
builder.append(state.getData()[i]);
if(i < length -1) {
builder.append("\t");
}
}
}
System.out.println(builder.toString());
} }
```

**Figure 5  An example of the main class code for data normalization.**

To prepare a set of data and its normalization a JAVA program was developed. Program randomly selects the values of the sensors and analytically calculates the expected values on the actuators. The generated data sets must be normalized so that the values are in the range [0..1]. For this purpose, at the 2nd stage, the program runs the developed normalization module and outputs the final results. The example of the main class code for data normalization is shown in Fig. 5.

An example of the generated training sample for training a neural network is shown in Fig. 6.

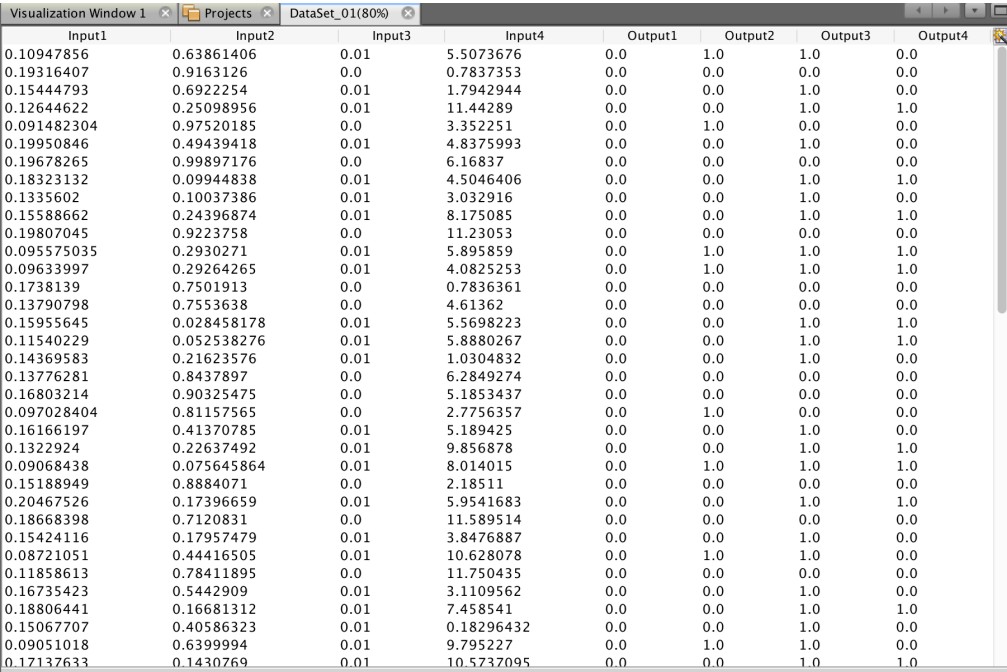

**Figure 6** A fragment of the training sample.

## Peculiarities of implementation and training of an artificial neural network

The NeurophStudio environment was used for designing, training and checking neural network correctness functioning. This is free software for various types of neural network design. It allows monitoring of the learning process, modifying the structure of the neural network, determining a set of training values, visualizing the learning results, *etc*.

A multilayer perceptron was chosen as a neural network and designed with the help of NeurophStudio. It consists of four input neurons and one balancing neuron, six internal neurons and one balancing neuron. The output layer contains four neurons.

The peculiarity of this artificial neural network is that all neurons of the first layer have a connection with all neurons of the second layer (except for the balancing one for the second layer). The situation is similar with the connections between the second and third layers, namely: seven neurons of the second layer have connections with all four neurons of the third layer (Fig. 7).

Within the environment, the structure and connections between neurons can be explored. For correct training of the neural network, it is recommended to divide the set into two parts. The first part is used to train the neural network, while the second part is a control set that can be used to test the neural network on new data. In this case, neural network training takes place on 80% of the data set.

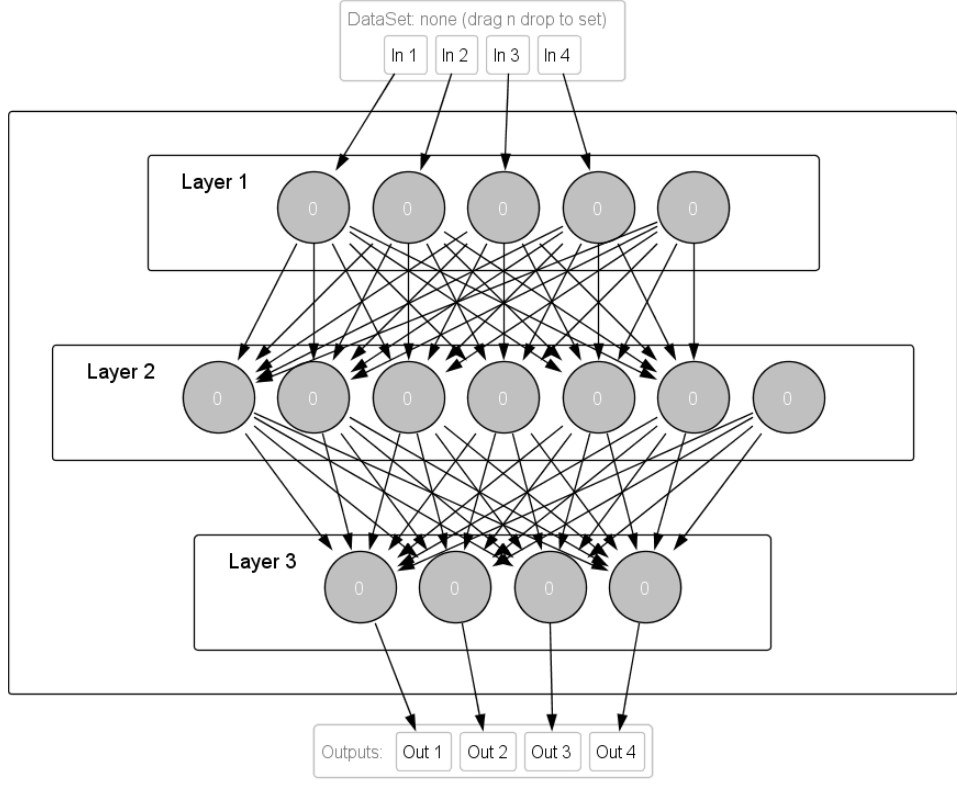

**Figure 7** Structure of the neural network of multilayer perceptron for the intelligent mini-greenhouse control system designed in NeurophStudio.

The training of neural network of multilayer perceptron is carried out by the backpropagation method. An example of the dependence of the neural network learning error on the number of iterations is shown in Figs. 8 and 9.

The neural network is tested on 20% of the data that was not involved in the training process, and the results of work on this sample provide an opportunity to check how the neural network functions on independent data.

Checking the accuracy of the functioning of the neural network for the intelligent mini-greenhouse control shows the mean squared error value of 0.032. The trained neural network is used for neuro-controller implementation.

# RESULTS AND DISCUSSION

## Hardware and software implementation of the neuro-controller

The development of a neuro-controller includes two main parts: hardware and software.

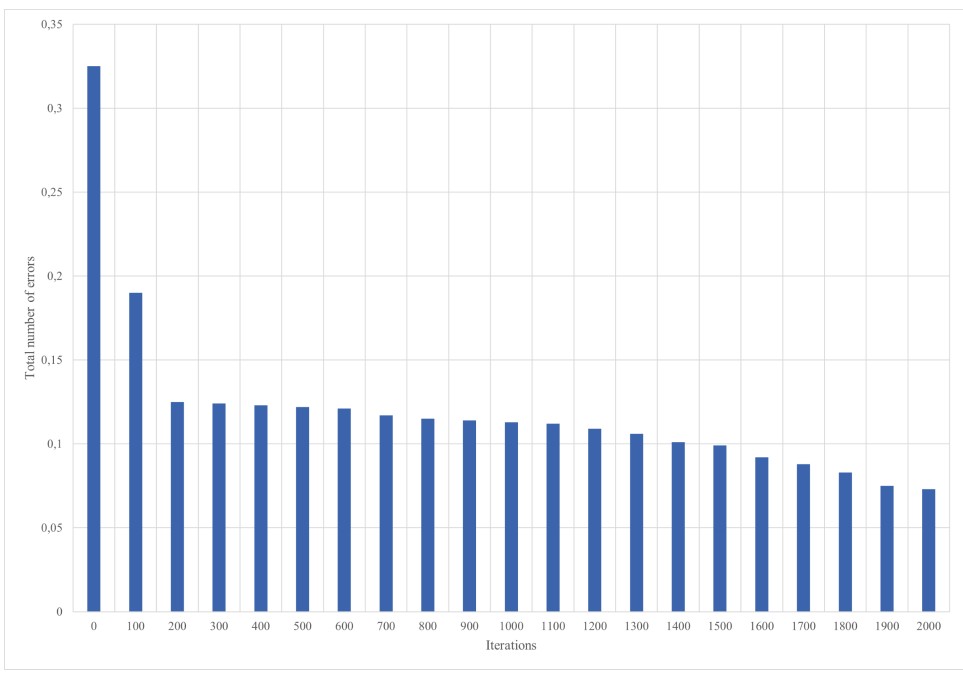

**Figure 8** **Results of neural network training, iterations 1–2,000.**

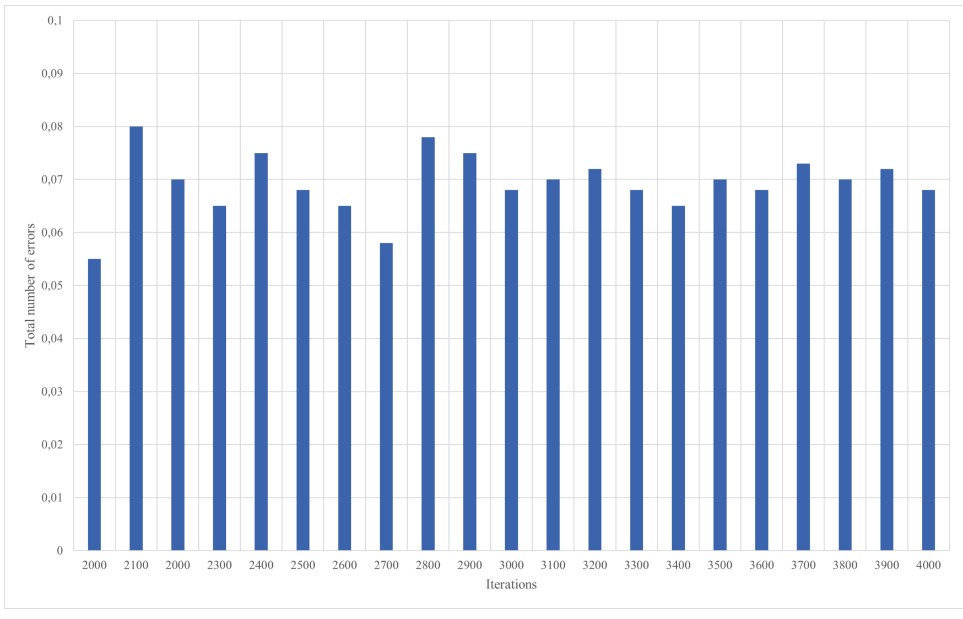

**Figure 9** **Results of neural network training, iterations 2,001–4,000.**

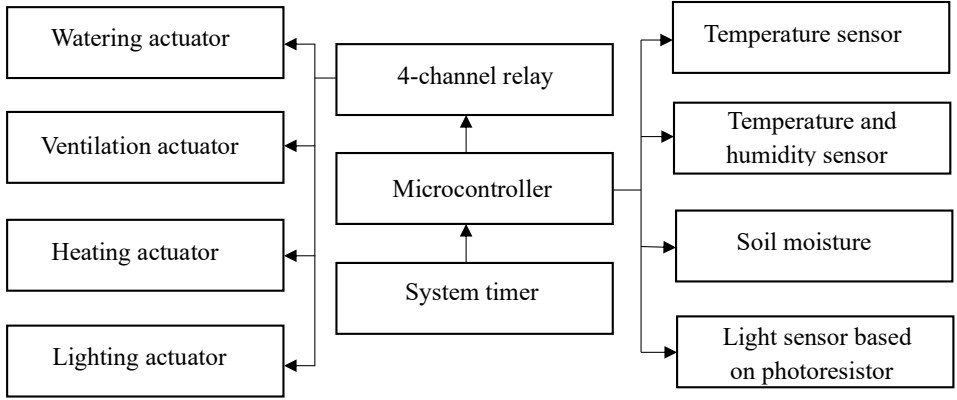

**Figure 10**  Structural diagram of the hardware of the intelligent mini-greenhouse control system.

## Features of the hardware implementation of the neuro-controller

We developed a structural diagram of the hardware part of the neuro-controller, which includes the following components (Fig. 10):

- STM32-F103C8T6 microcontroller, which is based on the ARM 32 Cortex-M3 MCU with an operating frequency 72 MHz, 64 KB of Flash memory and 20 KB of data memory;
- real time clock DS1307, which is necessary to determine the time of day;
- a number of sensors were used to monitor technological data on microclimate and lighting:
- temperature sensor DS18B20, which makes it possible to monitor the temperature in the range from −10 °C to 85 °C with an error of 0.5 °C;
- temperature and humidity sensor DHT11, which allows to monitor air humidity in the range from 20% to 80% with an error of 5%. Also, the sensor allows to monitor the temperature in the range from 0 °C to 60 °C with an error of 2%. The DS18B20 sensor gives a value with a smaller error, so it is recommended to use it in contrast to the DHT11;
- soil moisture sensor that measures soil conductivity and provides data in the range of 0–5 V depending on the set threshold value;
- photoresistor, on the basis of which the light sensor is organized and an auxiliary balancing resistor is used. The change in the voltage ratio between the balancing resistor and the photoresistor is determined by external lighting.
- relays are used as executive devices to control the power supply of the actuators.

The online EasyEDA development environment was used to design the hardware part of embedded control system for mini-greenhouse. This environment makes it possible to synthesize the electrical schematic diagram of the circuit board (Fig. 11).

The schematic diagram shows each component's pins and their connections.

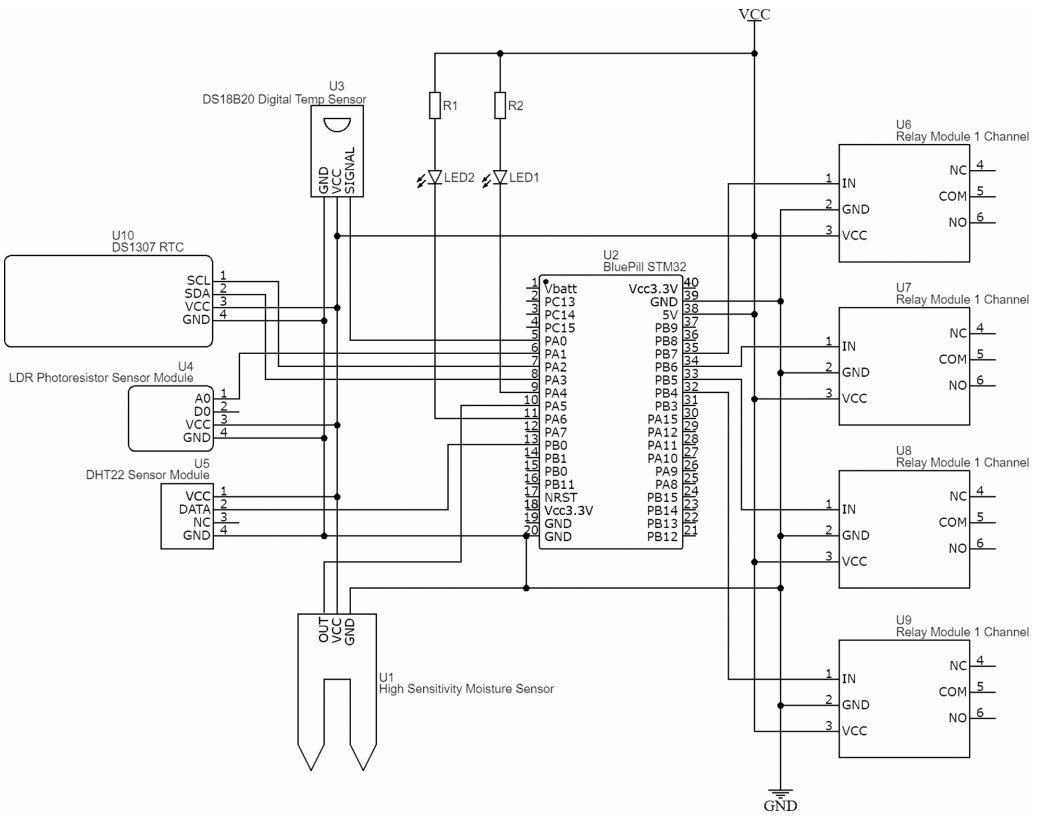

**Figure 11** Schematics of the hardware part of the embedded control system for mini-greenhouse.

## The software part of the developed embedded control system for mini-greenhouse

The software for an embedded control system for a mini-greenhouse consists of a set of modules. Each of them implements some operations, including the realization of the neuro-controller and its control. The interconnection of these modules is shown in Fig. 12. For system initialization, initialization of ports, sensors, actuators, real time clock and loading of initial data a system initialization module is responsible.

A sensor control module is used to periodically read data from sensors, and access the real time clock. It sets up the initial time and periodically reads the current time of day. The sensor control module is accessed by a periphery control module. This module also transmits control signals to the actuator control module by means of relays that change the state of the actuators.

The developed neuro-controller consists of several modules, described below. The neural network loading module is used for the neural network initialization, loading the neural network configuration, neuron function types, and weighting coefficients matrices. The neural network input/output module is responsible for loading input data into the neural network and reading the output data for the actuator's control. The Data processing module is used to coordinate the work of this modules. It also controls the Neural network emulation module. For emulation, the subroutine takes information about the type of

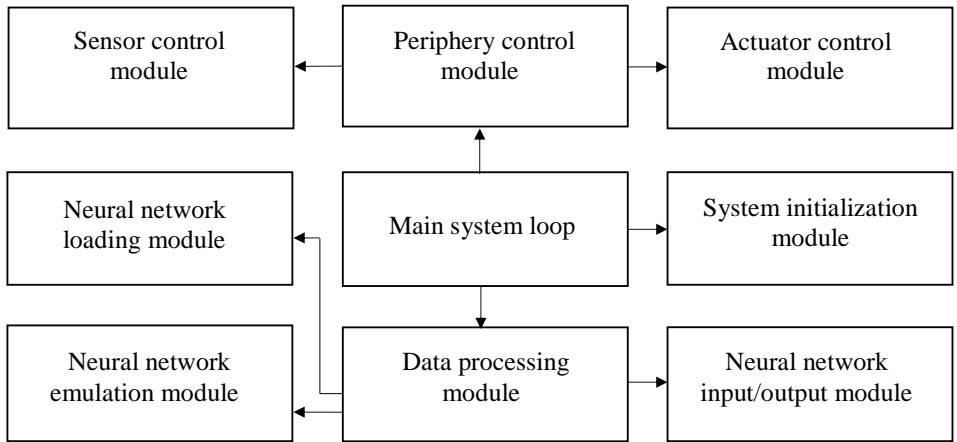

**Figure 12   The structure of the software part of an embedded control system for mini-greenhouse.**

neuron function, weighting coefficients, and connections between neurons and performs calculations of each neuron's value. As a result, the emulated neural network forms values that are used to control the mini greenhouse.

The main feature of the neuro-controller is its cyclical operation. Since there is no need to frequently poll and update the system state, the interrupt operator is used, which stops the main cycle of program execution and limits the frequency of execution of the main cycle.

During the execution of the main cycle, a subroutine is called to simulate the operation of the neural network. Based on its response, commands are formed to control the actuators.

The algorithm of emulating of the neural network operation is presented in Fig. 13.

The subroutine is responsible for calculating the values of each of the neurons. As input data of the neural network the measured values from sensors and the real time clock are used. At the output, the value of the state to which each of the actuators should be transferred (on/off) is calculated.

## CONCLUSIONS

The use of neural networks for applications in control systems has become increasingly popular. However, their implementation in embedded systems requires taking into account their limitations in performance, memory, *etc*. We proposed a neuro-controller for the embedded control system, which enables the processing of input technological data. It is based on the modular principle, which makes it possible to quickly modernize the technical system.

As a proof of concept, a neuro-controller for processing fuzzy and unstructured data for an intelligent mini-greenhouse based on an artificial neural network has been developed. It was conducted training of artificial neural network and error checking of the model, which does not exceed 3.2%.

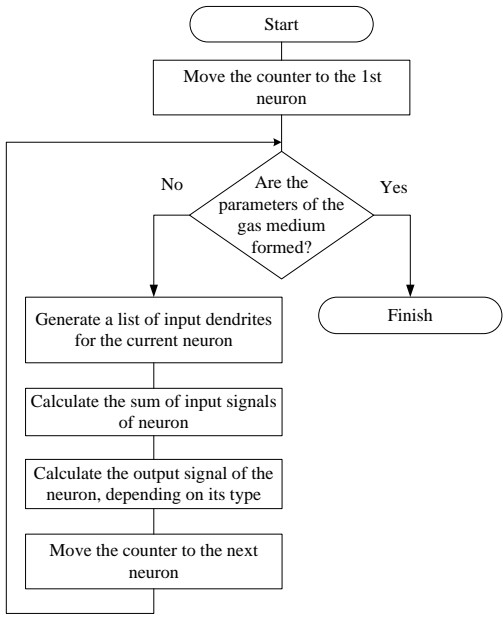

**Figure 13    Block diagram of the algorithm of the neural network emulation.**

The software and hardware of the neuro-controller for the intelligent mini-greenhouse was developed. The hardware is based on the STM32 microcontroller, which satisfy mass and size limitations for that type of device. The software for setting up the neural network is implemented in Java, and the system software of the microcontroller is implemented by standard development tools. In the future, a such neuro-controller can be used in other control systems. In this case neural network can be trained on another dataset to achieve its control functions.

### Funding
The authors received no funding for this work.

### Competing Interests
Natalia Kryvinska is an Academic Editor for PeerJ.

### Author Contributions
- Vasyl Teslyuk conceived and designed the experiments, performed the experiments, performed the computation work, authored or reviewed drafts of the article, and approved the final draft.
- Ivan Tsmots conceived and designed the experiments, performed the experiments, performed the computation work, authored or reviewed drafts of the article, and approved the final draft.

- Natalia Kryvinska conceived and designed the experiments, authored or reviewed drafts of the article, and approved the final draft.
- Taras Teslyuk conceived and designed the experiments, performed the experiments, analyzed the data, performed the computation work, prepared figures and/or tables, authored or reviewed drafts of the article, and approved the final draft.
- Yurii Opotyak performed the experiments, authored or reviewed drafts of the article, and approved the final draft.
- Mariana Seneta analyzed the data, prepared figures and/or tables, authored or reviewed drafts of the article, and approved the final draft.
- Roman Sydorenko analyzed the data, authored or reviewed drafts of the article, and approved the final draft.

### Data Availability

The raw data and computer code of the program for neural network training are available in the Supplementary Files.

### Supplemental Information

Supplemental information for this article can be found online at http://dx.doi.org/10.7717/peerj-cs.1680#supplemental-information.

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
