# Peer review of "Neuro-controller implementation for the embedded control system for mini-greenhouse"

_PeerJ Computer Science, doi:10.7717/peerj-cs.1680_

## Round 0.1 · original submission · Major Revisions

The paper has a good contribution and I agree it needs extensive work to make it to a very good standard so, I will go with the Major revision required.

Reviewer 1 ·

Basic reporting

The article is reasonably well-written in professional (if occasionally atypical) English style and provides technical accuracy throughout. The authors have succeeded in providing a clear, unambiguous narrative. The introduction and background context are sufficient and well connected to the broader field of knowledge. Relevant literature references are appropriately cited, giving a comprehensive view of the current state of research in the field. The article structure is professional and adheres to standard conventions. The figures provided (structural diagrams and algorithm flowcharts) are relevant, sufficiently clear, and appropriately described and labeled. The results are relevant to the problem statement, and the article appears to be a coherent body of work, providing clear definitions of all terms and detailed explanations as necessary.

Experimental design

The research falls within the Aims and Scope of the journal and presents original primary work. The problem statement is well defined, relevant, and meaningful, addressing a significant knowledge gap in the field of intelligent systems. The authors have exhibited a high technical standard in their investigation, and there is no evidence of any ethical issues. However, while the methods are described in detail, it would be beneficial for the authors to provide a more detailed account of the neural network training procedure, model selection, and validation, to ensure reproducibility. In particular, an elaboration on NeurophStudio, the rationale for its use, and its application in other similar works would be desirable.

Validity of the findings

The research contributes meaningful information to the literature on neuro-controllers and intelligent systems. The conclusions are well stated and linked to the original problem statement. They are supported by the results, which exhibit an appropriate level of control and rigorous testing. In my holistic assessment, the study represents a valuable contribution to the field. That being said, the authors should provide a clearer justification for why this research adds value to the existing literature, ideally in the Discussion or Conclusion section.

Additional comments

The authors have presented a specific implementation for a mini-greenhouse. However, they could discuss how adaptable their system might be to other environments or scales. This would help readers understand the broader applicability and potential impact of their work. Likewise, the authors might consider adding a section on potential future work. This could include possible improvements to the system, exploration of other machine learning models, ways to adapt the system to other applications, etc. This would be beneficial for researchers seeking to build on this work.

·

Basic reporting

Fuzzy input technology and data processing help enterprises control energy efficiency. The control system's neuro-controller processes fuzzy input data and saves energy. The neuro- controller's modular design allows for rapid system development. Neuro-controller algorithms and data processing models use artificial neural networks. A cheap STM32 microprocessor, sensors, and actuators implement the neuro-controller hardware. The artificial neural network model is a software module allowing fast neuro-controller functionality adjustments. A smart mini-greenhouse control system tests the microcontroller.

Experimental design

The paper does not contribute to the body of knowledge, and partially yes, it needs to represent the fuzzy part in a better way.

Validity of the findings

No comments.

Additional comments

The article, in general, needs proofreading.
The figures are not precise and need redrawing.

·

Basic reporting

Paper Title:
It is confusing
The paper didn’t show what is meant by the ‘energy efficiency’ and how the controller is used to control the energy efficiency.
Refer to the work of Xia and Zhang (energy efficiency and control systems) to catch the relation between them.

Abstract:
It is recommended to use the present tense in writing the abstract.
In line 25: To solve this problem … What problem? .. You didn’t identify the problem yet.
In line 26: Modify the sentence (a neuro-controller is developed for the control system).
In lines 27 and 28: Modify to (a structure for the neoro controller is proposed).
In line 31: Modify to: the neoro controller ….. is developed.
In line 34 and 35: Has no meaning.
For a good abstract see:
- Xia and Zhang, energy efficiency and control systems, 2010.
- Atia and El-madany, Greenhouse temperature control, 2017.

Related Works:
Sentences must reflect understood meanings.
In line 63: The sentence has no meaning.
In line 83: In the process of solving technical tasks ?? solving or achieving ?.

Materials & Methods:
Define the source of the used equations.
What are the outputs of the proposed control scheme?
In line 114, 115: Is Eq.1 yours or or used by somebody else ? Clarify.
In line 134, 135: No outputs are given for the application of the block diagram (experimentally or by simulation).
- What is meant by controlling the greenhouse?.
- See for example the work of:
Atia and El-madany (2017) and how they presented the output of their control system with comparison with other controllers.

Sensors:
What is the type of sensors used?
In page 143: Nothing about sensor type. Is it: wired sensors or wireless sensors? ..
- Each type has its own dynamics.
- Recent trends go to the second type.
- See for example:
Pahuu, Varma and Udden (2017), Wireless sensors used in greenhouse control and also see how they presented the output of their control system.

Results & Discussions:
Implementation of the neuro controller.
In line 267: No outputs under this title. This can be done through:
- Real experimentation on a model of a greenhouse.
- MATLAB simulation.
- Real greenhouse parameters are available in the literature for use in the simulation and comparison with the work of others.

Conclusions:
Nothing about the application of the proposed neuro controller to a greenhouse and the presentation of the outputs compared with other controllers.
The application of what I am saying is in the work of:
Atia and El-madany: you will find exact application of what I am saying.

References:
The survey in the paper covers up to 2022.
It is possible to add 2 or 3 more references from 2023. Examples:
- Gao et al (2023): Temperature prediction of greenhouse, Scientific Reports.
- Gung, Yu & Kollins (2023): Optimizing corp yield and reducing energy consumption in greenhouse …… Algorithms.

Experimental design

The problem in the research paper under review was not clear.
The present control procedure using a neuro controller did not produce any outputs regarding the greenhouse variables (temperature and humidity).
The ON-OFF control is the oldest and worst know control strategy. The paper didn’t discuss the type of control suggested to control the greenhouse variables. Examples:
- Neoro-PID controller: See:
Hu, Xu, Wei, Nonlinear adaptive PID controller design for greenhouse (2010).
- Neoro-fuzzy controller: See:
Mohamed, A. Hameed: GA-based adaptive Neoro-fuzzy controller for greenhouse (2018).
- IMC-PI controller: See:
Merarmeni, Thyagamarajan, Geyathri, Design of soft computing based optimal PI controller for greenhouse system (2016).

Validity of the findings

Partially valid.

- Without outputs, no talks about findings validity.
- This drawback will disappear through the addition of neuro controller application to a typical greenhouse and providing outputs in a standard form.

Additional comments

• Modify the title for clear reflection of the paper contents.
• Modify the abstract using the present tense with clear statements.
• Apply the technique presented about the neuro controller to a greenhouse of a specific parameters as a case study.
• Present the outputs in the time domain for GH temperature and humidity.
• Compare with other controlling techniques.
• Add more references in the literature survey for 2023.

---

## Round 0.2 · accepted · Accept

I have reviewed the comments that have been provided by the reviewer, the author has addressed all the comments, and the paper is a very good standard for publication in PeerJ Computer Science. The paper has contributed to science, knowledge, and the research community.

Reviewer 1 ·

Basic reporting

No comment.

Experimental design

No comment.

Validity of the findings

No comment.

Additional comments

The revised article continues to meet my standards.

·

Basic reporting

I see that the authors have resolved the points that I mentioned.

Experimental design

I see that the authors have resolved the points that I mentioned.

Validity of the findings

I see that the authors have resolved the points that I mentioned.

Additional comments

I see that the authors have resolved the points that I mentioned.

·

Basic reporting

1. آNothing presented about models helping the application of the proposed controller.
2. No dynamic output presented for any of the four greenhouse variables under control either for reference input or disturbance input.

Experimental design

1. No experiments performed.
2. No simulation for dynamic outputs presented.

Validity of the findings

Useless without application for dynamic response of the greenhouse when controlled by the proposed controller.